# Cholesterol Oxime Olesoxime Assessed as a Potential Ligand of Human Cholinesterases

**DOI:** 10.3390/biom14050588

**Published:** 2024-05-15

**Authors:** Dora Kolić, Goran Šinko, Ludovic Jean, Mourad Chioua, José Dias, José Marco-Contelles, Zrinka Kovarik

**Affiliations:** 1Division of Toxicology, Institute for Medical Research and Occupational Health, 10001 Zagreb, Croatia; dkolic@imi.hr (D.K.); gsinko@imi.hr (G.Š.); 2Université Paris Cité, CNRS, Inserm, CiTCoM, F-75006 Paris, France; ludovic.jean1@u-paris.fr; 3Institute of General Organic Chemistry (CSIC), 28006 Madrid, Spain; m.chioua@csic.es (M.C.); jlmarco@iqog.csic.es (J.M.-C.); 4Institut de Recherche Biomédicale des Armées, 91220 Brétigny-sur-Orge, Paris, France; jose.dias@def.gouv.fr; 5Faculty of Science, University of Zagreb, 10000 Zagreb, Croatia

**Keywords:** reactivation, organophosphate poisoning, warfare nerve agent, neuroprotection, neurodegeneration

## Abstract

Olesoxime, a cholesterol derivative with an oxime group, possesses the ability to cross the blood–brain barrier, and has demonstrated excellent safety and tolerability properties in clinical research. These characteristics indicate it may serve as a centrally active ligand of acetylcholinesterase (AChE) and butyrylcholinesterase (BChE), whose disruption of activity with organophosphate compounds (OP) leads to uncontrolled excitation and potentially life-threatening symptoms. To evaluate olesoxime as a binding ligand and reactivator of human AChE and BChE, we conducted *in vitro* kinetic studies with the active metabolite of insecticide parathion, paraoxon, and the warfare nerve agents sarin, cyclosarin, tabun, and VX. Our results showed that both enzymes possessed a binding affinity for olesoxime in the mid-micromolar range, higher than the antidotes in use (i.e., 2-PAM, HI-6, etc.). While olesoxime showed a weak ability to reactivate AChE, cyclosarin-inhibited BChE was reactivated with an overall reactivation rate constant comparable to that of standard oxime HI-6. Moreover, in combination with the oxime 2-PAM, the reactivation maximum increased by 10–30% for cyclosarin- and sarin-inhibited BChE. Molecular modeling revealed productive interactions between olesoxime and BChE, highlighting olesoxime as a potentially BChE-targeted therapy. Moreover, it might be added to OP poisoning treatment to increase the efficacy of BChE reactivation, and its cholesterol scaffold could provide a basis for the development of novel oxime antidotes.

## 1. Introduction

Olesoxime (cholest-4-en-3-one) is a small molecule compound that was first synthesized and evaluated in 2007 as a potential drug candidate for the treatment of the fatal neurodegenerative disorder, amyotrophic lateral sclerosis (ALS) [1]. Olesoxime is a cholesterol derivative, which is present as a stable mixture of *syn*- and *anti*-isomers of the oxime side-chain (RR’C=N–OH) in the 3-position of the compound [1]. It is highly lipophilic (cLogP = 10) and is able to cross the blood–brain barrier (BBB), although it has poor aqueous solubility at pH 7.4 and must be orally administered in an oily excipient [1,2,3]. Olesoxime was initially identified based on its survival-promoting activity on purified cultured rat motor neurons deprived of neurotrophic factors, as well as on striatal and cortical neurons under various stress conditions [1]. Further *in vivo* studies established its neuroprotective and therapeutic effect, leading to preclinical and clinical studies for ALS and spinal muscular atrophy (SMA) [2,3,4]. Although no significant benefits were observed in patients [5,6], olesoxime is still being investigated for the treatment of various neurodegenerative diseases due to its broad neuroprotective effect in different types of neurons. Similar to other ring-oxidized oxysterol compounds that modulate neurotransmission and have numerous biological activities in the nervous system [7,8,9], olesoxime inhibits the mitochondrial permeability transition pore (mPTP) complex, which mediates the mitochondrial cell death program via calcium and cytochrome c release [1], modulates oxidative stress and reactive oxygen species (ROS) production [4], regulates calcium and cholesterol homeostasis and improves mitochondrial respiration [3]. Due to its various modes of action, olesoxime demonstrates potential applicability for multiple neurodegenerative disorders, including Huntington’s disease [10] and Parkinson’s disease [11], and offers some benefits for treating Alzheimer’s disease [12].

Since olesoxime contains an oxime group (Figure 1), it could potentially act as an antidote in cases of organophosphorus compounds (OP) poisoning. OPs like pesticides and nerve agents (NA) are potent inhibitors of acetylcholinesterase (AChE) and butyrylcholinesterase (BChE) due to the phosphylation of their catalytic serine [13,14]. OPs exert their toxic effect primarily by inhibiting AChE, an essential enzyme for neurotransmitter acetylcholine (ACh) hydrolysis, which leads to the accumulation of ACh in synapses and overstimulation of cholinergic receptors. Both enzymes can be found in the synapses of the central nervous system, neuromuscular junctions of the peripheral nervous system, and in blood where AChE is bound to the erythrocyte membrane and BChE is dissolved in plasma [15]. Poisoning induces a plethora of symptoms like miosis, bronchorrhea, bradycardia, convulsions, and in severe poisoning cases loss of consciousness and respiratory failure, as well as long-term neurological damage in survivors [14,16]. Prevention of seizures and recovery of enzyme activity is the primary goal of therapy, which generally involves the administration of anticholinergic atropine and an oxime reactivator of phosphylated AChE [14]. Compounds with an oxime group can restore the activity of inhibited cholinesterases through the nucleophilic attack of the oximate anion on the phosphorus atom of the phosphylated catalytic serine of AChE and BChE [13]. So far, only three pyridinium oximes have been approved for military and clinical use: pralidoxime (2-PAM), asoxime (HI-6), and obidoxime [13].

In addition to not being equally effective reactivators for all OPs, standard pyridinium oximes are hydrophilic and cannot cross the BBB in sufficient concentrations for AChE reactivation due to the permanent charge on the nitrogen atom. Therefore, only 1–10% of the oximes’ plasma concentration is present in the brain, and their action is mostly limited to the peripheral nervous system [17]. Moreover, oximes can also be toxic at doses needed for reactivation [13] and their likely adverse toxic effects and risk-benefit ratio have to be considered in preclinical and clinical studies [18]. Since olesoxime displays excellent safety and tolerability properties and effectively crosses the BBB (reviewed in [3]), in this study we analyzed olesoxime *in vitro* as a ligand and potential reactivator of phosphylated AChE and BChE. In addition, we investigated the reactivation kinetics of the combined olesoxime and 2-PAM and determined the important interactions of olesoxime within the enzyme active site.

## 2. Materials and Methods

### 2.1. Chemicals for Cholinesterase Assays

Olesoxime was synthesized from commercial (+)-4-cholesten-3-one, as previously reported [19] in 95% yield, and isolated as a mixture of *E*/*Z* isomers in a 2/1 ratio that we did not attempt to separate, and submitted together for biological evaluation. The synthesized olesoxime showed spectroscopic and analytical data are in good agreement with its structure and the data described in the literature [19].

Olesoxime was dissolved in dimethyl sulfoxide (DMSO; Kemika, Zagreb, Croatia) as a 100 mM solution, diluted in methanol as a 10 mM solution, and stored at 4 °C. Oxime 2-PAM (Sigma-Aldrich, St. Louis, MO, USA) was prepared in water as a 100 mM solution. Nerve agents sarin, cyclosarin, VX, and tabun (NC Laboratory, Spiez, Switzerland) and insecticide paraoxon (Sigma-Aldrich, St. Louis, MO, USA) were prepared as stock solutions (5 g/L) in isopropyl alcohol and stored at 4 °C. Further dilutions were made in water just before use. Acetylthiocholine iodide (ATCh), thiol reagent 5,5′-dithiobis(2-nitrobenzoic acid) (DTNB), and bovine serum albumin (BSA) were purchased from Sigma-Aldrich (St. Louis, MO, USA). Stock solutions were prepared in water (ATCh) or 0.1 M sodium phosphate buffer, pH 7.4 (DTNB). The final concentration of ATCh in the IC_50_ determination was 0.1 mM, and 1.0 mM in all reactivation experiments. The final concentration of DTNB was 0.3 mM for all measurements.

Recombinant human AChE was diluted in 1% BSA phosphate buffer as the work solution. Human BChE was obtained from purified human plasma and was diluted in phosphate buffer. Enzymes were stored at 4 °C.

### 2.2. Determination of Oxime Inhibition Potency as IC_50_ Value

To determine the IC_50_ value, we measured enzyme activity in the presence of a wide range of oxime concentrations, ensuring maximum inhibition compared to the control activity. The assay was performed in the 96-well plates on the Tecan Infinite M200PRO plate reader. The inhibition mixture (300 μL final) contained phosphate buffer, AChE or BChE, olesoxime, and DTNB (0.3 mM final), and following the addition of ATCh (0.1 mM final), the activity was assayed by Ellman’s method [20]. The final concentrations of DMSO were kept under 0.15%, to eliminate their influence on enzyme activity [21]. The measured activity was corrected for oxime-induced hydrolysis of ATCh. The IC_50_ values were determined from at least three experiments by a nonlinear fit of the oxime concentration logarithm values vs. % of enzyme activity using Prism 9 (Graph Pad Software, San Diego, CA, USA).

### 2.3. In Vitro Reactivation Assay

For reactivation screening measurements, BChE or AChE was first incubated with paraoxon, sarin, cyclosarin, VX, or tabun for up to 1 h, achieving 95–100% inhibition. Inhibited enzymes were then filtered through Mobicol Spin G-50 columns (MoBiTec GmbH, Goettingen, Germany) to remove excess unconjugated OP. After filtration, the enzymes were diluted 10-fold in 0.1 M sodium phosphate buffer pH 7.4, additionally containing 0.01% BSA for AChE, and incubated with 0.1 mM olesoxime at 25 °C. At specified time intervals up to 24 h, an aliquot was taken and diluted 100-fold in buffer containing DTNB. The recovered enzyme activity was measured upon the addition of the substrate ATCh (1 mM) by the Ellman spectrophotometric method [20]. The uninhibited enzyme was identically diluted and passed through a parallel column, and control activity was measured in the presence of oxime at concentrations used for reactivation. Enzyme activity measurements were performed at 25 °C and 412 nm, on a CARY 300 spectrophotometer with a temperature controller (Varian Inc., Australia). The observed first-order rate constant of reactivation (*k*_obs_) was calculated by linear regression of the reactivation dependence (React./%) on time of reactivation (t/h).

For the detailed reactivation kinetics using a wider olesoxime concentration range (from 10 to 130 μM), and the combined reactivation assay with 2-PAM (0.1 or 1.0 mM) and 50 μM olesoxime, we determined reactivation constants *k*_2_ (maximal first-order reactivation rate constant), *K*_OX_ (apparent phosphylated enzyme-oxime dissociation constant) and *k*_r_ (overall second-order reactivation rate constant), as well as React_max_ (maximal reactivation percentage). *k*_2_ and *K*_OX_ were evaluated from the plot of the observed first-order rate constant of reactivation (*k*_obs_) vs olesoxime concentration, and *k*_r_ was the ratio of *k*_2_ and *K*_OX_ as previously described [22].

### 2.4. In Silico Molecular Modeling

Olesoxime to be docked in the active site of AChE and BChE was modeled and later minimized with the MMFF94 force field using ChemBio3D Ultra 12.0 (PerkinElmer, Inc., Waltham, MA, USA). The Discovery Studio 20.1 (BioVia, San Diego, CA, USA) Dock Ligands protocol (CDOCKER) with a CHARMm force field was used for the docking study [23,24]. As a model of AChE, we used the crystal structure of human AChE (PDB code 4PQE; [25]). The model of BChE was the crystal structure of human BChE (PDB code 2PM8; [26]). The binding site within the AChE and BChE was defined by a sphere (r = 13 Å) and it was used as the rigid receptor [27]. Details about the docking procedure using the CDOCKER protocol and scoring of generated ligand poses by a CHARMm energy were described previously [23].

## 3. Results and Discussion

Olesoxime is a cholesterol derivative with neuroprotective properties whose oxime group may provide antidotal activity in cases of OP compound poisoning. Olesoxime differs from most types of studied cholinesterase reactivators which are mostly pyridinium or imidazolium oximes containing a quaternary nitrogen atom [28,29,30,31,32,33,34,35,36,37] or small uncharged oximes with multiple ionization states [38,39,40]. Human cholinesterases differ in their binding affinity towards oximes and OP inhibitors alike, which is a direct consequence of the divergence in the active site gorge between AChE and BChE [22]. While AChE has 14 aromatic amino acids lining the gorge, BChE has 6 of them replaced with aliphatic amino acids, resulting in a 200 Å^3^ larger active site [41]. Therefore, bulkier molecules have a higher inhibition rate for BChE than for AChE due to the easier access to the catalytic gorge. This trend was also observed for olesoxime when we assessed the binding affinity of AChE and BChE for the compound in terms of IC_50_ value (Figure 2). Olesoxime was dissolved in DMSO and diluted as 10 mM in methanol to limit the inhibition of cholinesterases by DMSO in both inhibition and reactivation mixtures when using compound concentrations higher than 150 µM [21,42]. Methanol is a less potent inhibitor of the enzymes while improving the solubility of lipophilic compounds such as olesoxime in phosphate buffer solution. However, both enzymes displayed poor affinity for olesoxime in the lower micromolar range and its solubility was impaired at concentrations above 100 µM even with the added methanol. Total cholinesterase inhibition was not achieved and we approximated IC_50_ to be over 200 µM for AChE and around 100 µM for BChE, which is more than 3 orders of magnitude lower than that of tacrine [43]. Our results are consistent with other studies where bulkier molecules were generally more potent BChE inhibitors due to the difference in active site size [27,34,37,42,44].

Molecular modeling of the reversible complex of olesoxime within the active sites of the two cholinesterases enabled us to visualize interactions between the oxime compound and the amino acid residues lining the active site gorge (Figure 3). The oxime group of olesoxime faces the exit of the AChE active site gorge (Figure 3A) and many hydrophobic interactions with surrounding residues are present (purple). There is no interaction with Trp86 from the choline-binding site, an important residue for substrate hydrolysis, but there is interaction with His447 from the catalytic triad. Olesoxime also interacts with Trp286, a residue from the peripheral anionic substrate binding site. In BChE, however, olesoxime interacts with residues of the catalytic triad and choline-binding site, allowing the reactivation of OP-BChE conjugate and a higher binding affinity of BChE for the compound compared to AChE (Figure 3B). Moreover, the oxime group forms a hydrogen bond (green) with Glu197, which is located next to the catalytic Ser198, while hydrophobic interactions with His438 (catalytic triad) and Trp82 (choline substrate binding site) stabilize a productive orientation of olesoxime within the BChE active site.

It is worth emphasizing that although olesoxime exists as a stable mixture of syn- and anti-isomers at the 3-position, we did not prefer one of the isomers for the molecular modeling. The best-ranked poses resulted in the AChE complex with *syn*-isomer, and the BChE complex with *anti*-isomer at the 3-position (Figure 3). However, in accordance with our recent study on uncharged 2-thienostilbene oximes, AChE and BChE were inhibited similarly with pure isomers, and, therefore, we do not expect that preparation of pure olesoxime’s isomers would have merits on biological activity [45].

Olesoxime was next tested for its efficacy in reactivating AChE and BChE inhibited by the organophosphate insecticide paraoxon and the nerve agents cyclosarin, sarin, tabun and VX (Figure 1), and the results were expressed in terms of the observed reactivation rate constant (*k*_obs_) and maximal reactivation percentage (React_max_) achieved with OP-inhibited enzymes within 24 h (Figure 4). The 0.1 mM olesoxime exhibited substantial reactivation of cyclosarin-inhibited BChE, reaching about 60% of recovered enzyme activity within 3 h. Other OP-BChE conjugates were not susceptible to being reactivated by olesoxime, reaching <30% of their reactivation maximum. All tested OP-AChE conjugates also showed resistance to reactivation. It could be that the 0.1 mM concentration of the olesoxime was too low to ensure efficient oxime binding to the phosphylated BChE. However, higher concentrations could not be tested due to the aforementioned limitation in olesoxime solubility. In the case of AChE, the resistance to reactivation results from the low affinity of the enzyme for olesoxime in 0.1 mM concentration which was used in the screening assay. Furthermore, the affinity for the oxime is also limited by the spatial constraints within the active site gorge, which are determined by the structure and orientation of the organophosphate bound to the catalytic serine of the enzyme [22]. Additionally, the modeling results show that the oxime group is oriented toward the exit of the active site, precluding proper reactivation. Previous studies have shown that phosphoroamidate OPs like tabun are generally resistant to reactivation, probably due to the electron pair located on the amidic group making the nucleophilic attack by oxime very difficult [29,30,46]. Paraoxon-inhibited BChE conjugates are also not easily reactivated, and standard pyridinium oximes remain among the most effective POX-ChE antidotes [34,47,48], with some studies demonstrating the *in vitro* and *in vivo* potential of edrophonium-based oxime [49], and novel zwitterionic oximes [38,39,40,50]. Pyridinium oximes are generally less effective reactivators of OP-BChE conjugates than OP-AChE, and efforts are directed toward the synthesis of imidazolium, cinchona, and quinuclidinium scaffold-based oximes to find more potent OP-BChE reactivators [34,37,51].

Based on the screening results, only the cyclosarin-BChE complex was selected for further study of reactivation kinetics (Figure 5). A reactivation assay was performed over a wider concentration range (from 10 to 130 μM) and about 70% of maximum reactivation was achieved within 3 h. Due to a lack of a saturation curve, the overall reactivation rate constant (*k*_r_) for cyclosarin-inhibited BChE was determined as the slope of dependence of *k*_obs_ vs. olesoxime concentration, and it was in the range of the standard oxime HI-6 and higher than oxime 2-PAM (Table 1). While olesoxime did not show an improved recovery of enzyme activity compared to standard and some novel quinuclidinium oximes [37], it showed a reactivation efficiency of the cyclosarin-BChE conjugate in the range of some di-chlorinated pyridinium-based oximes [35,36] and 3-hydroxy-2-pyridine aldoximes [21], which were synthesized with the premise of improving the overall lipophilicity of charged pyridinium molecules and stimulating passive diffusion across the BBB. Although some of these oximes were predicted to efficiently cross the BBB based on both *in silico* predictions and the *in vitro* brain membrane permeability test, olesoxime has the advantage of being a highly lipophilic compound that is known to successfully cross the BBB and it was tested in clinical studies that proved its safety in patients [2,3,4]. Due to these properties, its cholesterol scaffold could serve as a model for the further development of novel oxime antidotes. Moreover, as olesoxime provides mitochondria-stabilizing, calcium-modulating, and anti-apoptotic effects [3], it may improve oxidative status and attenuate signs of inflammation, which are characteristic consequences of OP poisoning in survivors [16]. In addition, olesoxime is a ligand of a presumptive mMPT pore regulator, the 18 kD a mitochondrial permeability transition-translocator protein (TSPO), and as such can play an important role in neuroprotection both by modulating the endogenous production of neurosteroids in the nervous system, and by regulating the MPT process [52,53].

Since olesoxime is well tolerated and was clinically tested without side effects, we tested the combined reactivation efficacy of olesoxime and 2-PAM pair on sarin- and cyclosarin-BChE conjugates (Figure 6). The combination of 50 µM olesoxime and 0.1 or 1.0 mM 2-PAM in the reactivation mixture increased React_max_ by 10–30% for both sarin- and cyclosarin-inhibited BChE and increased the observed rate constant of reactivation (*k*_obs_) for the cyclosarin-BChE complex. The beneficial effect of olesoxime was especially noticeable for the cyclosarin-BChE complex and 0.1 mM 2-PAM concentration, where the addition of 50 µM olesoxime increased the reactivation maximum between 30 and 40%. A conceptually similar *in vivo* study in rats poisoned with sarin by Caisberger et al. [54] concluded that combinations of the oximes TMB-4 or K203 with HI-6 did not interfere with HI-6 bioavailability, but rather supported its action. Therefore, combining the oximes in the antidotal treatment may be a promising step towards a broad-spectrum antidotal treatment of acute nerve agent exposure. A similar study in a guinea pig model [55] evaluated the efficacy of a combined therapy of the standard oximes obidoxime and 2-PAM treatment regimen compared to a double dose of either oxime alone or a single equimolar equivalent dose during sarin poisoning. Combined oxime therapy resulted in improved seizure control, increased peripheral and central cholinesterase reactivation, and improved behavioral signs. Although olesoxime is not an AChE reactivator, its peripheral action might assist BChE in neutralizing sarin or cyclosarin in the blood before they reach target tissues, thereby establishing a bioscavenging enzyme-oxime pair and preventing the development of long-lasting neurological damage [37,48,49].

## 4. Conclusions

We evaluated the capacity of the cholesterol-based oxime olesoxime to perform as an antidote and centrally active reactivator for organophosphate insecticide- and nerve agent-inhibited cholinesterase enzymes, AChE, and BChE. Cholinesterases displayed weak binding affinities for olesoxime, which is a consequence of its large cholesterol scaffold. Olesoxime showed substantial reactivation efficacy for the cyclosarin-BChE conjugate in the standard pyridinium oxime HI-6 range.

## 5. Future Directions

Since olesoxime has the advantage of successfully crossing BBB over charged pyridinium oximes, olesoxime might be added to OP poisoning treatment to increase the efficacy of dephosphylation of nerve agent-inhibited BChE in combination with standard oximes, while its cholesterol scaffold could provide a basis for the development of novel oxime antidotes. Indeed, in counteracting the OP poisoning, developing multitarget drugs such as olesoxime might be of particular interest to enable simultaneous activity on AChE/BChE reactivation ensuring the neurotransmission and mitochondrial calcium-dependent functions to provide an overall neuroprotective effect.

## Figures and Tables

**Figure 1 biomolecules-14-00588-f001:**
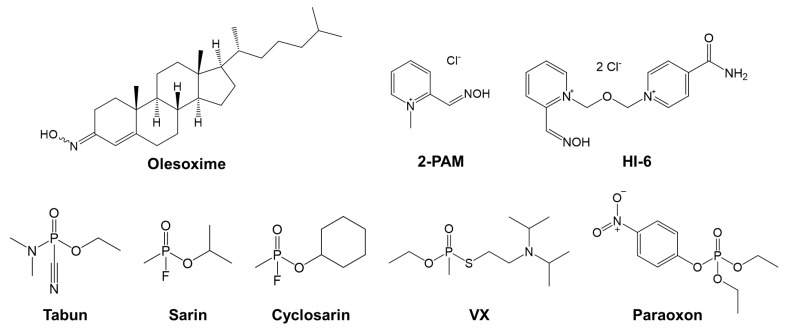
Chemical structures of olesoxime, standard oxime reactivators of AChE—pralidoxime (2-PAM) and asoxime (HI-6), and organophosphate compounds (tabun, sarin, cyclosarin, VX, and paraoxon).

**Figure 2 biomolecules-14-00588-f002:**
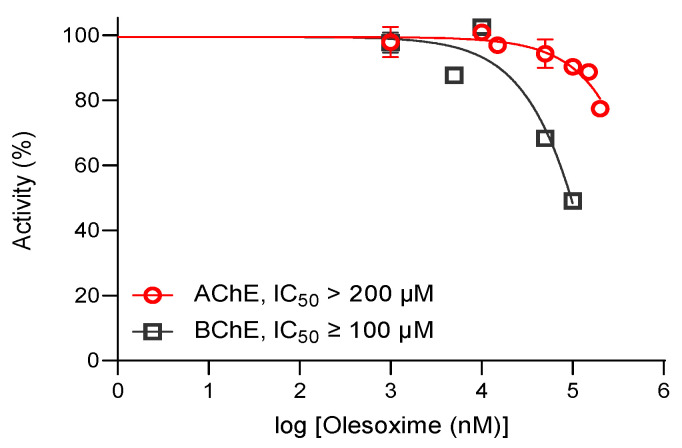
The inhibition profile and estimated binding affinities of AChE and BChE for olesoxime in terms of IC_50_ (±SEM) values measured at 25 °C. Total inhibition was not achieved due to olesoxime’s low solubility in buffer solution at concentrations higher than 100 µM.

**Figure 3 biomolecules-14-00588-f003:**
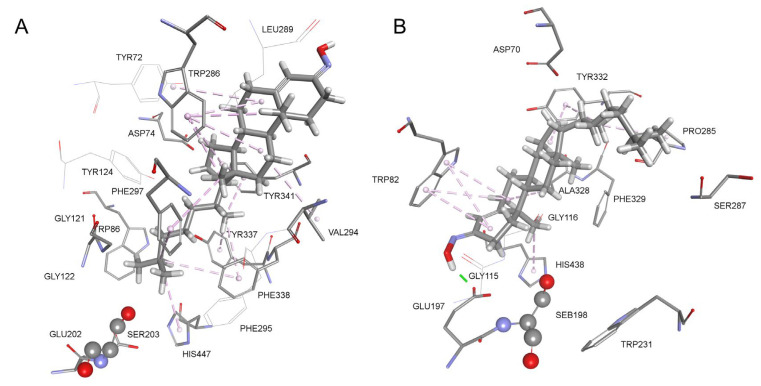
Complex between native human AChE (**A**) and BChE (**B**) with olesoxime. Interactions with amino acid residues are represented as dashed lines: hydrophobic (purple) and hydrogen bond (green). Crystal structure of human AChE (PDB code 4PQE) [25] and human BChE was used (PDB code 2PM8) [26].

**Figure 4 biomolecules-14-00588-f004:**
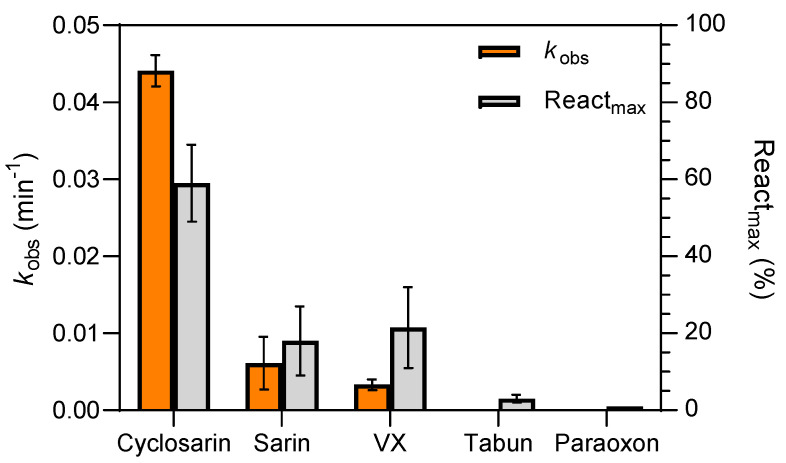
Reactivation screening of nerve agent-inhibited BChE by 0.1 mM olesoxime (±SEM) expressed in terms of the observed reactivation rate constant (*k*_obs_) and maximum reactivation percentage (React_max_) determined at 25 °C within 24 h (n = 2).

**Figure 5 biomolecules-14-00588-f005:**
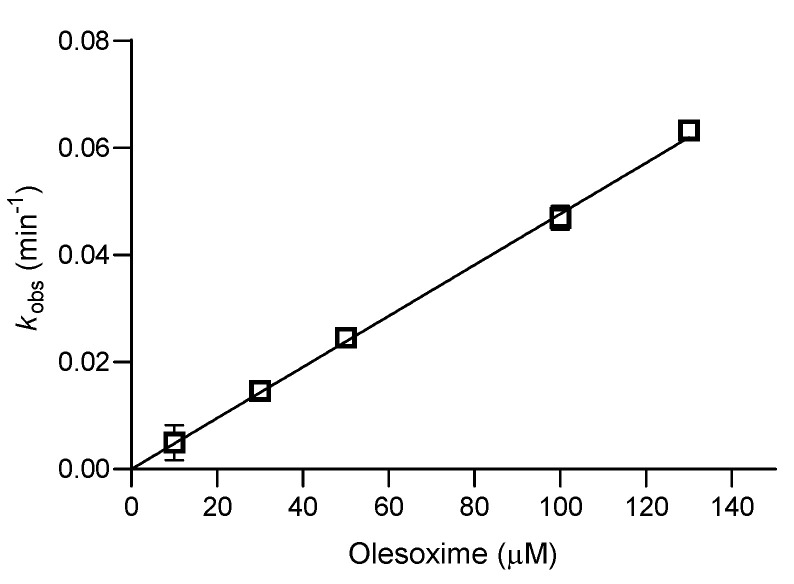
Kinetics of olesoxime-assisted reactivation of cyclosarin-inhibited BChE (±SEM) determined at 25 °C within 4 h (n = 2).

**Figure 6 biomolecules-14-00588-f006:**
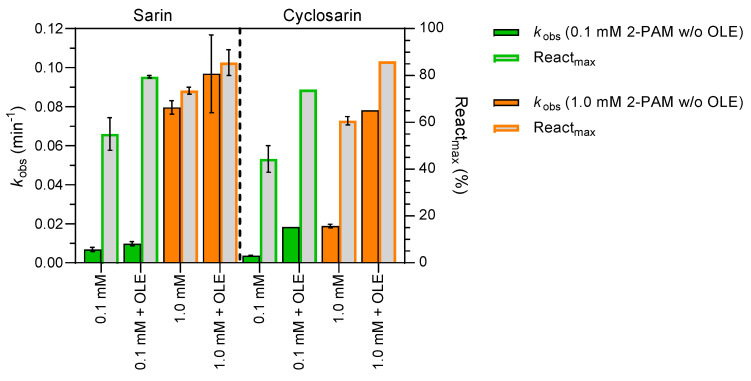
Combined reactivation capability of 0.1 or 1.0 mM oxime 2-PAM with or without 50 µM olesoxime (OLE) on sarin- and cyclosarin-BChE conjugates (±SEM). The observed reactivation rate constant (*k*_obs_) and maximum reactivation percentage (React_max_) were determined at 25 °C within 24 h (n = 2).

**Table 1 biomolecules-14-00588-t001:** Reactivation of cyclosarin-inhibited BChE by olesoxime and standard oximes 2-PAM and HI-6. The kinetic parameters (±SEM): first-order reactivation rate constant (*k*_2_), phosphylated enzyme-oxime dissociation constant (*K*_OX_), the second-order reactivation rate constant (*k*_r_), reactivation maximum (React_max_) and time of reaching the reactivation maximum (t) were determined at 25 °C within 24 h.

Oxime	*k*_2_/min^−1^	*K*_OX_/µM	*k*_r_/M^−1^ min^−1^	React_max_/%	t/h
Olesoxime *	/	/	477 ± 12	70	3
2-PAM [35]	0.08 ± 0.01	1200 ± 290	65 ± 10	80	1
HI-6 * [21]	/	/	780 ± 30	90	0.5

* Linear dependence of *k*_obs_ vs. oxime concentration in the studied concentration range enabled to determine *k*_r_.

## Data Availability

Data are available upon request.

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
