# Peer review of "Cholesterol Oxime Olesoxime Assessed as a Potential Ligand of Human Cholinesterases"

_biomolecules, 2024, doi:10.3390/biom14050588_

Round 1
Reviewer 1 Report
Comments and Suggestions for Authors
This is a clearly written manuscript describing some impressive and useful data sets. My only two concerns were the following: 1) paraoxon is not an insecticide, it is the active metabolite of the insecticide parathion, so the wording should be changed to be an accurate description; 2) some of the concentrations tested in the in vitro tests were extremely and unrealistically high, for example, 130 micromolar; these should be justified.
Author Response
Thank you for the positive report.
In revised manuscript we described paraoxon as the active metabolite of the insecticide parathion.
With all respect to the comment on the concentration in the in vitro tests that were extremely and unrealistically high, concentration of 130 micromolar compound is in a standard concentration range for reactivation assay e.g. it is not rare that concentration of reactivator is high as 1 mM.
Moreover, there is no a saturation curve (Figure 5) as obtained by 2-PAM, and therefore we add a half sentence (page 7) about that phenomenon - showing that 130 uM concentration in reactivation is not high. The limitation for testing even with higher concentration was its poor solubility in buffer.
Reviewer 2 Report
Comments and Suggestions for Authors
Kovarik et al. presented well designed and conducted study on reactivation of nerve agents and insecticide inhibited ChEs by the olesoxime. Olesoxime was carefully chosen compound for this research. Although no significant success regarding reactivation of ChEs was obtained, disscussion is very well presented so I think it will be valuable and interesting to the readers of Biomolecules and to the scientists in this field.
I cannot find anything that required additional work or description. The only concern in about potential influence of geometry of tested compound on the results. Since a mixture of E/Z isomers was used for biological evaluation, do authors think that separation and biological evaluation of pure isomers would be useful? Can the obtained model (Figure 3A and 3B) help in choosing a suitable geometry for each enzyme?
Part Result and discussion: concentration of olesoxime is expressed in 0.1 mM (lines 213 and 221) and 100 µM (line 217). Choose one way.
Author Response
We thank for the favorable critiques and we accepted all suggestions.
In our recent paper (https://doi.org/10.3390/ph14111147) we reported that inhibition of BChE with pure E/Z isomers were similar. Similarly we do not expect that separation of preparation pure isomers would have merits for further biological assessment. Concerning the modelling, our software uses both isomers. One paragraph on discussion on isomers is added to the manuscript, page 6.
We uniformed the olesoxime concentration to 0.1 mM.
Reviewer 3 Report
Comments and Suggestions for Authors
In this manuscript, Kolić et al tried to present their work of olesoxime as an antidote and centrally active reactivator for organophosphate insecticide- and nerve agent-inhibited cholinesterase enzymes, AChE, and BChE, providing a basis for the development of novel oxime antidotes. It may be suitable for publication in this journal after revisions: there are some errors and irregular writings, for examples, “olesoxime demonstrated potential applicability for multiple neurodegenerative disorders, including Huntington’s disease, and Parkinson’s disease, and offers some benefits for treating Alzheimer’s disease ”, the verb tense of this sentence is inconsistent; The horizontal coordinate of Figure 2 is incorrect; What’s the meaning of “the cholesterol-based oxime olesoxime”?
Comments on the Quality of English Languagemoderate
Author Response
We agree with comments and corrected errors in writing: "demonstrated" is replaced with "demonstrates" and "cholesterol-based" is replaced with "cholesterol derivative".
Concerning the X-axe, Figure 2 - we confirm that it is correct - e.g. log 5 = 10,000 nM = 100 μM that approximates IC50 of BChE.